# Idiopathic Plasmacytic Lymphadenopathy Forms an Independent Subtype of Idiopathic Multicentric Castleman Disease

**DOI:** 10.3390/ijms231810301

**Published:** 2022-09-07

**Authors:** Asami Nishikori, Midori Filiz Nishimura, Yoshito Nishimura, Fumio Otsuka, Kanna Maehama, Kumiko Ohsawa, Shuji Momose, Naoya Nakamura, Yasuharu Sato

**Affiliations:** 1Department of Molecular Hematopathology, Okayama University Graduate School of Health Sciences, Okayama 700-8558, Japan; 2Department of Pathology, Okayama University Hospital, Okayama 700-8558, Japan; 3Department of General Medicine, Okayama University Graduate School of Medicine, Dentistry, and Pharmaceutical Sciences, Okayama 700-8558, Japan; 4Department of Medicine, John A. Burns School of Medicine, University of Hawai’i, Honolulu, HI 96813, USA; 5Department of Pathology, Saitama Medical Center, Saitama Medical University, Saitama 350-8550, Japan; 6Department of Pathology, Tokai University School of Medicine, Kanagawa 259-1193, Japan

**Keywords:** Castleman disease, idiopathic multicentric Castleman disease, idiopathic plasmacytic lymphadenopathy, plasma cell morphology

## Abstract

Idiopathic multicentric Castleman disease (iMCD) is a type of Castleman disease that is not related to KSHV/HHV8 infection. Currently, iMCD is classified into iMCD-TAFRO (thrombocytopenia, anasarca, fever, reticulin fibrosis, and organomegaly) and iMCD-NOS (not otherwise specified). The former has been established as a relatively homogeneous disease unit that has been recently re-defined, while the latter is considered to be a heterogeneous disease that could be further divided into several subtypes. In 1980, Mori et al. proposed the concept of idiopathic plasmacytic lymphadenopathy (IPL), a disease presenting with polyclonal hypergammaglobulinemia and a sheet-like proliferation of mature plasma cells in the lymph nodes. Some researchers consider IPL to be a part of iMCD-NOS, although it has not been clearly defined to date. This is the first paper to analyze iMCD-NOS clinicopathologically, to examine whether IPL forms a uniform disease unit in iMCD. Histologically, the IPL group showed prominent plasmacytosis and the hyperplasia of germinal centers, while the non-IPL group showed prominent vascularity. Clinically, the IPL group showed significant thrombocytosis and elevated serum IgG levels compared to the non-IPL group (*p* = 0.007, *p* < 0.001, respectively). Pleural effusion and ascites were less common in the IPL group (*p* < 0.001). The IPL group was more likely to have an indolent clinical course and a good response to the anti-IL-6 receptor antibody, while the non-IPL counterpart frequently required more aggressive medical interventions. Thus, the IPL group is a clinicopathologically uniform entity that forms an independent subtype of iMCD.

## 1. Introduction

Castleman disease (CD) is a rare lymphoproliferative disorder described by Castleman et al. in 1956 [1]. CD is clinically classified into unicentric and multicentric types. Unicentric CD (UCD) is characterized by a localized lymphadenopathy with or without minimal systemic symptoms, and the resection of the affected lymph node is often curative [2]. In contrast, multicentric CD (MCD) shows a generalized lymphadenopathy with systemic inflammatory symptoms, such as generalized weakness and fever [3]. The infection status of Kaposi sarcoma-associated herpesvirus/Human herpesvirus type 8 (KSHV/HHV8) defines the etiology of MCD [4]. Idiopathic MCD (iMCD) is defined as a group of KSHV/HHV8-negative MCD without POEMS syndrome (polyneuropathy, organomegaly, endocrinopathy, M-proteins, and skin changes) [5]. Clinically, iMCD is classified into iMCD-TAFRO (thrombocytopenia, anasarca, fever, reticulin fibrosis, and organomegaly) [6,7,8] and iMCD-NOS (not otherwise specified). Histologically, there are two main pathological variants in iMCD: plasma cell (PC) and hypervascular (HyperV) types [5,9,10]. The mixed type shows the features of both the PC and HyperV variants, but no clear pathological definition [11,12]. Commonly, iMCD-TAFRO is histologically associated with the HyperV type, and iMCD-NOS frequently has PC morphology [5]. As the name suggests, iMCD-NOS is a heterogenous entity. Previous studies have suggested that iMCD-NOS could include atypical and undiagnosed autoimmune diseases [13]. Moreover, it could potentially be further classified into several subtypes with research efforts, including clinicopathological analyses and genomic sequencing [14,15]. One potential candidate that is to be separated from the current iMCD-NOS is idiopathic plasmacytic lymphadenopathy (IPL). IPL was initially proposed in 1980 by Mori et al., characterized by polyclonal hypergammaglobulinemia and a sheet-like proliferation of mature plasma cells in the lymph nodes, as well as the exclusion of known diseases associated with hypergammaglobulinemias such as infections, collagen diseases, hyperthyroidism, allergic diseases, hepatitis, liver cirrhosis, and lymphoma [16]. In the clinical course, IPL was indolent, and all the cases achieved the remission of disease activity [13,16]. The concept of IPL was proposed before the establishment of MCD, and IPL was later considered as a part of iMCD-NOS, given the clinicopathological similarity [11,16]. However, there has been no study to validate whether or not IPL has distinct clinicopathologic features compared to other iMCD-NOS. In this study, we perform a comprehensive clinicopathological analysis of iMCD-NOS, with a focus on IPL or others (non-IPL) to examine if IPL needs to be defined as an independent iMCD subtype.

## 2. Results

### 2.1. Clinical Findings

The main clinical findings are summarized in Table 1. See Appendix A for details of each case.

Of the 42 included cases, 34 (81.0%) and 8 (19.0%) were classified as the IPL group and the non-IPL group, respectively. In the IPL group, there were 21 males and 13 females, aged 34–76 years, with a median age of 54.8. In the non-IPL group, three were males and five were females, aged 32–89 years, with a median age of 57.9. There was no significant difference in age between the two groups (*p* = 0.471). Regarding laboratory findings, there were no significant differences in the white blood cells (WBC) nor the C-reactive protein (CRP) between the two groups (*p* = 0.200, *p* = 0.391, respectively). While hemoglobin (Hb) was considerably lower in the non-IPL group compared to the IPL group, there was no significant difference (*p* = 0.080). By contrast, platelet count (Plt) and serum immunoglobulin G (IgG) were significantly higher in the IPL group (*p* = 0.007, *p* < 0.001, respectively). All cases had elevated serum interleukin-6 (IL-6), but there was no significant difference between the two groups (*p* = 0.149). In total, 6/26 (23.1%) in the IPL group and 3/6 (50.0%) in the non-IPL group had disease-specific autoantibodies (*p* = 0.420). In the IPL group, the following specific autoantibodies were detected: myeloperoxidase-anti-neutrophil cytoplasmic antibodies (MPO-ANCA) (3/6, 50.0%), proteinase-3-anti-neutrophil cytoplasmic antibodies (PR3-ANCA) (2/6, 33.3%), anti-double-strand DNA antibody (ds-DNA) (2/6, 33.3%), anti-single-strand DNA antibody (ss-DNA) (1/6, 16.7%), anti-ribonucleoprotein antibody (RNP) (1/6, 16.7%), anti-cardiolipin antibody (1/6, 16.7%), and anti-mitochondrial M2 antibody (AMA2) (1/6, 16.7%). In contrast, the following specific autoantibodies were detected in the non-IPL group: anti-SS-A antibodies (SS-A) (2/3, 66.7%), anti-cardiolipin antibody (2/3, 66.7%), anti-platelet-associated IgG (PA-IgG) (2/3, 66.7%), ds-DNA (1/3, 33.3%), AMA2 (1/3, 33.3%), MPO-ANCA (1/3, 33.3%), and anti-SS-B antibodies (SS-B) (1/3, 33.3%). Regarding imaging, 5/8 (62.5%) in the non-IPL group had pleural effusions or/and ascites, while this was only noted in 1/34 (2.9%) in the IPL group (*p* < 0.001).

### 2.2. Pathological Findings

The pathological findings of the two groups are summarized in Table 2.

In the IPL group, various levels of vascularity were noted with a median score of 0.70, including 17/34 with grade 0 (50.0%), 11/34 with grade 1 (32.4%), 4/34 with grade 2 (11.8%), and 2/34 with grade 3 (5.9%). The median level of plasmacytosis was 2.9, including 3/34 with grade 2 (8.8%) and 31/34 with grade 3 (91.2%). The median score of regressed GCs was 0.90, including 8/34 with grade 0 (23.5%), 20/34 with grade 1 (58.8%), and 6/34 with grade 2 (17.6%). Regarding hyperplastic GCs, the median score was 2.4, including 1/34 with grade 0 (2.9%), 6/34 with grade 1 (17.6%), 7/34 with grade 2 (20.6%), and 20/34 with grade 3 (58.8%). The typical histological findings of the IPL group are shown in Figure 1.

Among the non-IPL group, the median level of vascularity was 2.0, including 1/8 with grade 0 (12.5%), 2/8 with grade 1 (25.0%), 1/8 with grade 2 (12.5%), and 4/8 with grade 3 (50.0%), respectively. In addition, the median score of the plasmacytosis was 2.1, including 1/8 with grade 1 (12.5%), 5/8 with grade 2 (62.5%), and 2/8 with grade 3 (25.0%), respectively. The median score of the regressed GCs was 1.4, including 2/8 with grade 0 (25.0%), 3/8 with grade 1 (37.5%), 1/8 with grade 2 (12.5%) and 2/8 with grade 3 (25.0%). The hyperplastic GCs showed a median score of 0.90, including 5/8 with grade 0 (62.5%), 2/8 with grade 2 (25.0%), and 1/8 with grade 3 (12.5%). The histological features of the non-IPL group are shown in Figure 2.

The IPL group had less significant vascularity, more frequent hyperplastic GCs, and severe plasmacytosis compared to the non-IPL group (*p* = 0.003, *p* = 0.003, *p* = 0.001, respectively). While the non-IPL group was more likely to have extensive regressed GCs, there was no significant difference between the two groups (*p* = 0.255). In addition, branching and/or whirl-like patterns of the vessels (Figure 2C–E) were observed in the GCs of the cases with prominent vascularization. Such findings were seen in 1/34 cases of the IPL group (2.9%) and 2/8 cases of the non-IPL group (25.0%).

### 2.3. Treatment and Clinical Course

Outpatient follow-up data were available for 23/34 cases in the IPL group and for 5/8 cases in the non-IPL group, with a median follow-up period of 65.5 months (Table 3).

For the first-line treatment, corticosteroids were used in 13/23 (56.5%) cases in the IPL group and 4/5 (80%) cases in the non-IPL group. One patient in the non-IPL group (Case 1) received corticosteroid and tocilizumab. Among those treated with corticosteroids, corticosteroids were successfully tapered in 3/13 (23.1%) patients in the IPL group and 1/4 (25.0%) patients in the non-IPL group. One case in the IPL group (Case 11) expired during the follow-up period with corticosteroid monotherapy, likely due to lung cancer.

Overall, 14/23 (60.9%) patients in the IPL group and 1/5 (20.0%) patients in the non-IPL group received tocilizumab during the follow-up period. Those in the IPL group who received tocilizumab achieved an improvement in disease activity. One patient in the IPL group treated with tocilizumab expired due to post-surgical bleeding that was unrelated to IPL (Case 15). In contrast, the non-IPL case treated with tocilizumab had a progressive disease and required rituximab as a second-line therapy to achieve a partial remission of disease activity.

## 3. Discussion

iMCD is a rare lymphoproliferative disorder that is characterized by multiple lymphadenopathies with unknown etiology [17]. In particular, iMCD-NOS is a heterogenous entity, likely including undefined disease [11,18,19,20,21]. The present results show that IPL is likely to be a separate subtype of iMCD, along with iMCD-TAFRO and iMCD-NOS, given its unique clinicopathological characteristics.

Our results show that the IPL group had distinct clinicopathological features compared to the non-IPL iMCD-NOS cases. Pathologically, the IPL group had less significant vascularity, as well as more prominent plasmacytosis and hyperplastic GCs than the non-IPL group. By contrast, the non-IPL group showed marked hypervascularization both in GCs and in interfollicular areas. Clinically, the IPL group had higher platelet counts and serum IgG levels, and fewer signs of fluid retention in third space such as pleural effusions and/or ascites, than the non-IPL group.

In 2008, before the current iMCD criteria were proposed, Kojima et al. reported that iMCD had at least two clinical subtypes, IPL and non-IPL, with the latter showing more thrombocytopenia, fluid retention, positive autoantibodies, and relatively aggressive clinical symptoms [13]. They also suggested that non-IPL may be associated with autoimmune diseases. In addition, Frizzera et al. reported multiple lesions of CD, which led to the term MCD being established [3,22]. Some of their MCD cases included those with clinical and laboratory findings characteristic of systemic lupus erythematosus (SLE), Sjögren’s syndrome, or both [3]. Currently, such cases may be considered ill-defined autoimmune diseases [23,24,25]. Moreover, SLE cases with MCD-like histology [24,26] and iMCD cases with various autoantibodies [23] have been reported. Although no significant differences were observed for disease-specific autoantibodies in the present results, this may be due to a lack of power to detect the difference, given the small number of non-IPL cases. Combined with the context and the present results, it may be crucial to closely follow-up with non-IPL patients on an outpatient basis to find clinical signs of autoimmune diseases.

The two groups also had different clinical courses and treatment responses. There were a few patients who were treated with tocilizumab (an anti-IL-6 receptor monoclonal antibody approved in Japan for the treatment of iMCD [27]), and all patients with IPL who received tocilizumab achieved a remission of disease activity. In contrast, the non-IPL case that had a poor response to tocilizumab also required rituximab. Despite the second-line non-IL-6 therapy, the patient still had progressive disease during the follow-up period. The results concur with previous studies suggesting that IPL may have an indolent clinical course compared to the non-IPL group, with a superior response to anti-IL-6 agents. In recent studies, the PI3K/Akt/mTOR pathway, JAK/STAT3 pathway, and type I IFN have focused on the treatment targets in iMCD cases refractory to IL-6-targeted therapy [28,29,30,31]. While non-IPL iMCD-NOS cases could be heterogenous, as discussed, efforts to identify a primary etiology (for example, possible autoimmune disease) by molecular analysis and targeted therapies for the cellular signals may need to be considered.

In conclusion, the present results suggest that IPL is clinicopathologically a uniform disease entity, and may be an independent subtype of iMCD. Future studies are warranted to identify diagnostics, treatment, and follow-up plans that are specific to IPL. Given the heterogeneity of the non-IPL cases, clinicians are urged to identify a primary etiology of such cases, including atypical autoimmune diseases. These cases may benefit from molecular analysis to clarify underlying pathology.

## 4. Materials and Methods

### 4.1. Patients

Forty-two Japanese patients with lymph node involvement of iMCD-NOS were included in this study. All cases were selected from pathology consultation files at the Department of Pathology, Okayama University. All patients had systemic or multiple lymphadenopathies, and the detailed sites are summarized in Appendix A. All iMCD-NOS patients met the consensus diagnostic criteria for iMCD [5]. All cases were serologically or immunohistochemically negative for KSHV/HHV8.

### 4.2. Histological Evaluation

All lymph node specimens were fixed in 10% formalin and embedded in paraffin. Paraffin-embedded tissue blocks were sliced into 3 µm thin sections and stained with hematoxylin and eosin (H&E), immunohistochemical staining and Berlin blue staining.

Immunohistochemical staining was performed using an automated BOND-III instrument (Leica Biosystems, Wetzlar, Germany) with the primary antibody of HHV-8 (13B10, 1:40; LifeSpan Biosciences, Seattle, WA, USA), CD138 (MI15, 1:200; DAKO, Carpinteria, CA, USA) and α-SMA (1A4, 1:50; DAKO, Carpinteria, CA, USA). In situ hybridization was also performed for the κ and λ light chains (Leica Biosystems, Wetzlar, Germany).

The pathological findings of all cases were reviewed with H&E stained by the authors (Y.S., M.F.N. and A.N.). The following histopathological features were graded on a scale of 0 to 3: vascularity, plasmacytosis, regressed germinal centers (GCs), and hyperplastic GCs, based on previous reports [5,28].

### 4.3. Classification of iMCD-NOS

According to a previous report [13], we defined IPL as a case meeting all the following four criteria: (1) prominent polyclonal hypergammaglobulinemia (γ-globulin > 4.0 g/dL or serum IgG level >3500 mg/dL), (2) multicentric lymphadenopathy, (3) an absence of definite autoimmune disease, and (4) normal germinal centers and a sheet-like infiltration of polyclonal plasma cells in the lymph node lesion. Cases that did not meet the criteria were defined as non-IPL.

### 4.4. Statistical Analysis

Statistical analyses were conducted using SPSS for Windows version 23.0 (SPSS, Chicago, IL, USA). Statistical significance was set at *p* < 0.05.

## Figures and Tables

**Figure 1 ijms-23-10301-f001:**
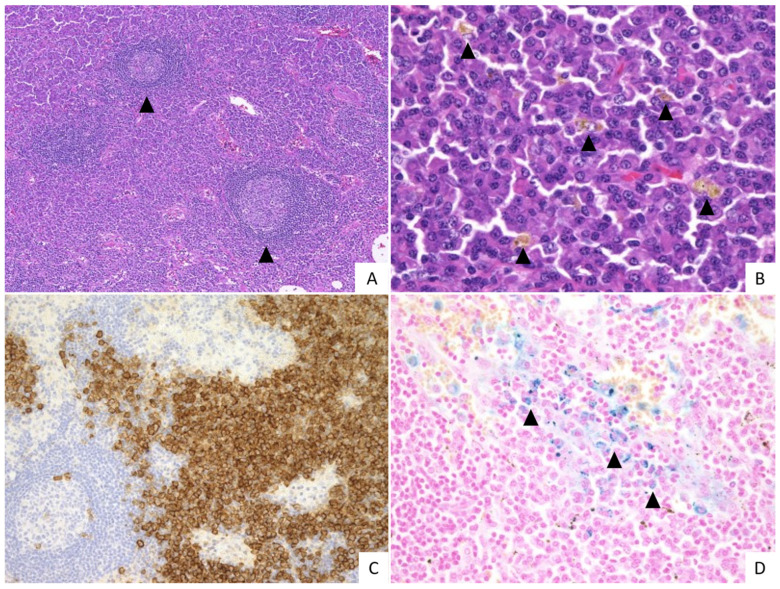
Histopathological features of the IPL group. (**A**) Interfollicular areas are expanded and germinal centers (GCs) appear hyperplastic (arrowheads) (H&E, 10×). (**B**) The sheet-like proliferation of mature plasma cells in the interfollicular areas and hemosiderin deposition are observed (arrow heads) (H&E,40×). (**C**) Numerous plasma cells are observed in the interfollicular areas (CD138, 20×). (**D**) Hemosiderin deposition is observed (arrowheads) (Berlin blue staining, 40×). This case was scored: vascularity grade 0, plasmacytosis grade 3, regressed GC grade 2, and hyperplastic GC grade 2.

**Figure 2 ijms-23-10301-f002:**
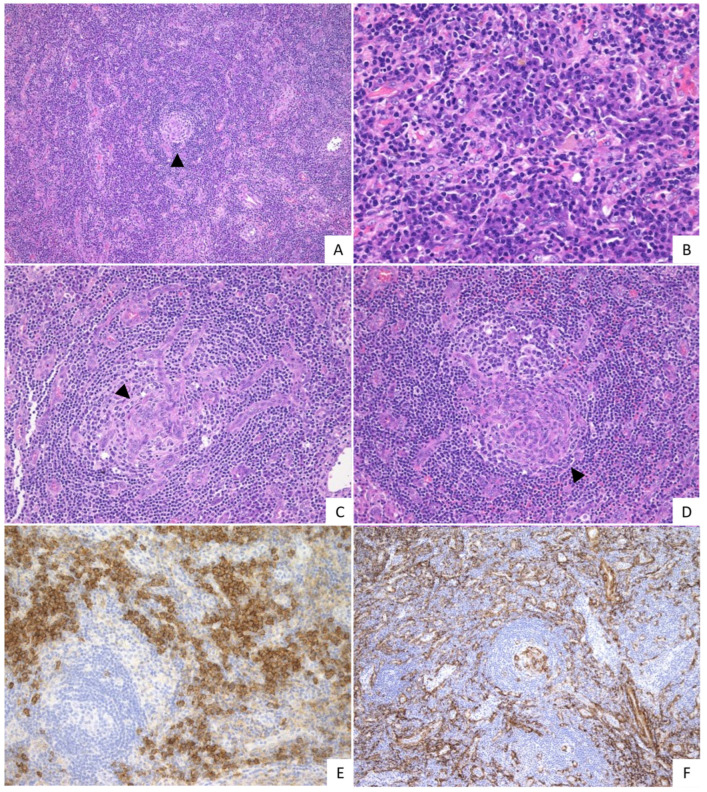
Histopathological features of the non-IPL group. (**A**) Extended interfollicular areas and severe vascularization are observed. Germinal center (GC) is atrophic (arrowhead) (H&E, 10×). (**B**) Mature plasma cells are observed within prominent vascularization (H&E, 40×). (**C**,**D**) Marked hypervascularization penetrating the GC is observed. The blood vessels show a branching pattern ((**C**), arrowhead) and whirl-like pattern ((**D**), arrowhead) (H&E, 20×). (**E**) Numerous plasma cells are observed in the interfollicular areas (CD138, 20×). (**F**) Vascular in the GC and interfollicular areas were α-SMA-positive (α-SMA, 10×). This case was scored vascularity grade 3, plasmacytosis grade 2, regressed GC grade 3 and hyperplastic GC grade 0.

**Table 1 ijms-23-10301-t001:** Clinicopathological findings of iMCD-NOS.

	IPL (*n* = 34)	Non-IPL (*n* = 8)	*p*-Value
Age (median ± SD)	54.8 ± 12.2	57.9 ± 16.9	0.471
Sex (M/F)	21/13	3/5	
WBC (×10³/µL)	7.7 ± 2.5 ^†^	14.1 ± 9.1 ^‡^	0.200
CRP (mg/dL)	6.5 ± 3.5 ^†^	13.2 ± 8.4 ^‡^	0.391
Hb (g/dL)	10.1 ± 2.1 ^†^	9.3 ± 1.6	0.080
Plt (×10⁴/µL)	36.9 ± 15.2 ^†^	23.8 ± 10.9 ^‡^	**0.007 ***
Serum IgG (mg/dL)	5140.3 ± 1453.1	2502.0 ± 752.3	**<0.001 ****
Serum IL-6 (pg/mL)	27.3 ± 16.8 ^†^	107.2 ± 94.2 ^‡^	0.149
Pleural effusions or/and ascites (%)	1 (2.9)	5 (62.5)	**<0.001 ****
Disease-specific autoantibody (%)	6/26 (23.1)	3/6 (50.0)	0.420

Significant *p*-values are in bold. Significance was calculated using the Mann–Whitney U test. Fisher’s exact analysis or chi-square test were used for the statistical analysis of nominal scales. * *p* < 0.05, ** *p* < 0.001. iMCD-NOS, idiopathic multicentric Castleman disease not otherwise specified; SD, standard deviation; IPL, idiopathic plasmacytic lymphadenopathy; WBC, white blood cells; CRP, C-reactive protein; Hb, hemoglobin; Plt, platelet; Ig, immunoglobulin; IL-6, interleukin 6. Normal ranges: WBC, 3.9–9.8 × 10³/µL; CRP, 0.0–0.3 mg/dL; Hb, 13.5–17.6 g/dL (male), 11.3–15.2 g/dL (female); Plt, 13.0–36.9 × 10⁴/µL; serum IgG, 870–1700 mg/dL; serum IL-6, 0.0–4.0 pg/mL. ^†^ WBC, CRP, Hb, Plt, and IL-6 levels were available for 21, 33, 31, 31, and 11 patients with IPL, respectively. ^‡^ WBC, CRP, Plt, and IL-6 levels were available for 7, 7, 7, and 5 patients with non-IPL, respectively.

**Table 2 ijms-23-10301-t002:** Pathological findings of iMCD-NOS.

	IPL (*n* = 34)	Non-IPL (*n* = 8)	*p*-Value
Vascularity			
Median	0.7	2.0	**0.003 ***
Grade 0 (%)	17 (50.0)	1 (12.5)	
Grade 1 (%)	11 (32.4)	2 (25.0)	
Grade 2 (%)	4 (11.8)	1 (12.5)	
Grade 3 (%)	2 (5.9)	4 (50.0)	
Plasmacytosis			
Median	2.9	2.1	**0.001 ***
Grade 0 (%)	0 (0.0)	0 (0.0)	
Grade 1 (%)	0 (0.0)	1 (12.5)	
Grade 2 (%)	3 (8.8)	5 (62.5)	
Grade 3 (%)	31 (91.2)	2 (25.0)	
Regressed GCs			
Median	0.9	1.4	0.255
Grade 0 (%)	8 (23.5)	2 (25.0)	
Grade 1 (%)	20 (58.8)	3 (37.5)	
Grade 2 (%)	6 (17.6)	1 (12.5)	
Grade 3 (%)	0 (0.0)	2 (25.0)	
Hyperplastic GCs			
Median	2.4	0.9	**0.003 ***
Grade 0 (%)	1 (2.9)	5 (62.5)	
Grade 1 (%)	6 (17.6)	0 (0.0)	
Grade 2 (%)	7 (20.6)	2 (25.0)	
Grade 3 (%)	20 (58.8)	1 (12.5)	

Significant *p*-values are in bold. Significance was calculated using the Mann–Whitney U test. * *p* < 0.05. GCs, germinal centers.

**Table 3 ijms-23-10301-t003:** Treatment and clinical courses of iMCD-NOS.

Subtype	Case No.	Age/Sex	1st Treatment	2nd Treatment	3rd Treatment	Outcome	Follow-Up Period (Month)
IPL	1	35/F	PSL 20 mg	rituximab	tocilizumab	Improved	122
	2	62/M	PSL 30 mg	tocilizumab		Improved	93
	3	55/F	tocilizumab			Improved	57
	4	37/M	follow-up			- ^†^	4
	5	39/M	tocilizumab			Improved	39
	6	49/F	tocilizumab			Improved	38
	7	59/M	PSL 50 mg			no response	39
	8	64/M	PSL 30 mg	tocilizumab		Improved	130
	9	70/M	PSL 30 mg	tocilizumab		Improved	32
	10	54/F	follow-up			no change	161
	11	65/M	PSL 25 mg	PSL 6 mg		Improved ^‡^	37
	12	62/F	PSL 10 mg	PSL 5 mg		Improved	175
	13	55/F	tocilizumab			Improved	13
	14	74/F	tocilizumab			Improved	10
	15	52/F	tocilizumab			Improved ^‡^	202
	16	70/M	follow-up			no change	198
	17	43/M	PSL 15 mg	tocilizumab		Improved	131
	18	41/M	PSL 5 mg			no response	126
	19	49/M	PSL 30 mg			Improved	90
	20	48/F	PSL 15 mg	tocilizumab		Improved	77
	21	76/M	PSL 40 mg	PSL 10 mg		Improved	6
	22	67/M	PSL 20 mg	tocilizumab		Improved	4
	23	72/M	tocilizumab			Improved	4
non-IPL	1	52/F	PSL 50 mg + tocilizumab	rituximab		PR	96
	2	89/M	PSL 60 mg			Repeatedly worsened during tapering	10
	3	73/F	PSL 35 mg			Improved	74
	4	49/F	follow-up			progression	41
	5	49/F	mPSL 500 mg	PSL 40 mg		no response	143

Treatment information was available for 23 cases of the IPL group and 5 cases of the non-IPL group. PR, partial remission; mPSL, methylprednisolone; PSL, prednisolone. “Outcome” represents the clinical condition of the patient at the last visit. “Partial remission” represents improvement in some laboratory data or subjective symptoms. “Complete remission” represents improvement in all laboratory data, objective symptoms, and radiographic findings. “No response” represents all clinical findings and subjective symptoms unchanged. “No change” represents no worsening nor improving of the disease during follow-up. “Progression” represents a worsening of laboratory findings, subjective symptoms, or radiographic findings. † Case 4: Lost to follow-up. ‡ Cases 11 and 15 achieved PR, but expired from non-iMCD disease (lung cancer) and post-surgical bleeding, respectively.

## Data Availability

Not applicable.

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
