# Peer review of "Idiopathic Plasmacytic Lymphadenopathy Forms an Independent Subtype of Idiopathic Multicentric Castleman Disease"

_ijms, 2022, doi:10.3390/ijms231810301_

Round 1

Reviewer 1 Report

Authors present results, which suggest that IPL is clinicopathologically an uniform disease entity, and may be an independent subtype of iMCD. This analysis is very interesting, and I  have only one reservation that the references contain a lot of works from years ago. However, the paper is valuable and I accept it in this form.

Author Response

Comment from Reviewer 1

  • Comment 1: [Authors present results, which suggest that IPL is clinicopathologically an uniform disease entity, and may be an independent subtype of iMCD. This analysis is very interesting, and I have only one reservation that the references contain a lot of works from years ago. However, the paper is valuable and I accept it in this form.]

Response: Thank you for reviewing our manuscript. We have revised the manuscript according to your suggestion by citing some recently-published articles listed below.

[15] Endo, Y.; Koga, T.; Ubara, Y.; Sumiyoshi, R.; Furukawa, K.; Kawakami, A. Mediterranean fever gene variants modify clinical phenotypes of idiopathic multi-centric Castleman disease. Clin Exp Immunol 2021, 206, 91-98, doi:10.1111/cei.13632.

[21] Nishimura, M.F.; Igawa, T.; Gion, Y.; Tomita, S.; Inoue, D.; Izumozaki, A.; Ubara, Y.; Nishimura, Y.; Yoshino, T.; Sato, Y. Pulmonary Manifestations of Plasma Cell Type Idiopathic Multicentric Castleman Disease: A Clinicopathological Study in Comparison with IgG4-Related Disease. J Pers Med 2020, 10, doi:10.3390/jpm10040269.

Reviewer 2 Report

Please use correct formatting.

Lines

54. use hyalinevascular/hypervascular at least the first time since hyalinevascular is more commonly used.

Table 1 and 2. use capital letters as indicated. For example Age...Grade...

Author Response

Comments from Reviewer 2

  • Comment 1: [Please use correct formatting.]

Response: Thank you for reviewing our manuscript. We have corrected formatting according to the journal’s requirements.

  • Comment 2: [Lines 54. use hyalinevascular/hypervascular at least the first time since hyalinevascular is more commonly used.]

Response: Thank you the comment. We are hoping to keep the sentence as is due to the following reason. Recently, researchers began to think the hyalinevascular and the hypervascular are different histologic types. The hyalinevascular type is a commonly-seen histology in unicentric Castleman disease, but rarely seen in iMCD.

  • Comment 3: [Table 1 and 2. use capital letters as indicated. For example Age...Grade...]

Response: We have capitalized the letters as suggested.

Reviewer 3 Report

Figure 1. Histopathological features of the IPL group. (A) Interfollicular areas are expanded and germinal centers (GCs) shows hyperplastic (HE, 10×). (B) The sheet-like proliferation of mature plasma cells in the interfollicular areas and hemosiderin deposition are observed. (HE,40×). This case was scored vascularity grade 0, plasmacytosis grade 3, regressed GCs grade 2, and hyperplastic GCs grade 2.

Comments:

please indicate with an arrow or a box the expansion of the germinal centers in A and the plasma cell elements in B and the accumulation of hemosiderin pigment in B. If you can also make a high magnification insert to document the cytological detail of the plasma cell elements because they are not clear to the observer of the images.

If you can show an immunohistochemical image for plasma cell elements and a histochemical image for iron (Pearls stain). This way the work will be better.

Please correct HE with H&E also in the text

Figure 2. Histopathological features of the non-IPL. (A) Extended interfollicular areas and severe vascularization are observed (HE, 10×). (B) Germinal centers (GCs) are atrophic, and high vascularity is observed in the GCs and interfollicular areas (HE, 20×). (C, D, E) Marked hypervascularization penetrating the GC is observed. Branching pattern(C) and whirl like pattern (D, E) are shown (HE, 20×). (F) Mature plasma cells are observed within prominent vascularization (HE, 40×). 151 This case was scored vascularity grade 3, plasmacytosis grade 2, regressed GCs grade 3 and hyperplastic GCs grade 0.

Comments:

please indicate with an arrow or a box as in figure 1.

If you can show an immunohistochemical image for plasma cell elements (CD138 etc. and CD34 or CD31 etc. for del vassels and others elements. This way the work will be better.

You must indicate the immunohistochemical markers you have made and show the images.

HHV-8 239 (13B10, 1:40; LifeSpan Biosciences, Seattle, USA). In situ hybridization was also performed 240 for the κ and λ light chains (Leica Biosystems, Wetzlar, Germany)……

Please correct HE with H&E also in the text

These changes are fundamental.

Thanks and good job

Author Response

Comments from Reviewer 3

  • Comment 1: [Figure 1. Histopathological features of the IPL group. (A) Interfollicular areas are expanded and germinal centers (GCs) shows hyperplastic (HE, 10×). (B) The sheet-like proliferation of mature plasma cells in the interfollicular areas and hemosiderin deposition are observed. (HE,40×). This case was scored vascularity grade 0, plasmacytosis grade 3, regressed GCs grade 2, and hyperplastic GCs grade 2.

Comments:

please indicate with an arrow or a box the expansion of the germinal centers in A and the plasma cell elements in B and the accumulation of hemosiderin pigment in B. If you can also make a high magnification insert to document the cytological detail of the plasma cell elements because they are not clear to the observer of the images.

If you can show an immunohistochemical image for plasma cell elements and a histochemical image for iron (Pearls stain). This way the work will be better.

Please correct HE with H&E also in the text]

Response: Thank you for reviewing our manuscript. According to your comment, we have indicated the hyperplastic GCs and hemosiderin deposition with arrows and added a high magnification figure. Additionally, we have added CD138 immunostaining and Berlin blue iron staining images to the figure 1. We also corrected HE to H&E throughout the manuscript.

  • Comment 2: [Figure 2. Histopathological features of the non-IPL. (A) Extended interfollicular areas and severe vascularization are observed (HE, 10×). (B) Germinal centers (GCs) are atrophic, and high vascularity is observed in the GCs and interfollicular areas (HE, 20×). (C, D, E) Marked hypervascularization penetrating the GC is observed. Branching pattern(C) and whirl like pattern (D, E) are shown (HE, 20×). (F) Mature plasma cells are observed within prominent vascularization (HE, 40×). 151 This case was scored vascularity grade 3, plasmacytosis grade 2, regressed GCs grade 3 and hyperplastic GCs grade 0.

Comments:

please indicate with an arrow or a box as in figure 1.

If you can show an immunohistochemical image for plasma cell elements (CD138 etc. and CD34 or CD31 etc. for del vassels and others elements. This way the work will be better.

You must indicate the immunohistochemical markers you have made and show the images.

HHV-8 239 (13B10, 1:40; LifeSpan Biosciences, Seattle, USA). In situ hybridization was also performed 240 for the κ and λ light chains (Leica Biosystems, Wetzlar, Germany)……

Please correct HE with H&E also in the text

These changes are fundamental.

Thanks and good job]

Response: According to your comment, we indicated the atrophic GCs with arrows. In addition, we added images for CD 138 and α-SMA immunostainings to the figure 2. Additional details of the immunohistochemical staining markers have been noted in Materials and Methods. As above, we corrected HE to H&E throughout the manuscript.
